# Subsurface nickel boosts the low-temperature performance of a boron oxide overlayer in propane oxidative dehydrogenation

Xiaofeng Gao[1,2,6], Ling Zhu[3,6], Feng Yang [ORCID][4,6], Lei Zhang [ORCID][4], Wenhao Xu[1], Xian Zhou[1], Yongkang Huang[1], Houhong Song[1], Lili Lin [ORCID][5], Xiaodong Wen [ORCID][3] ✉, Ding Ma [ORCID][2] ✉ & Siyu Yao [ORCID][1] ✉

Oxidative dehydrogenation of propane is a promising technology for the preparation of propene. Boron-based nonmetal catalysts exhibit remarkable selectivity toward propene and limit the generation of $CO_x$ byproducts due to unique radical-mediated C−H activation. However, due to the high barrier of O·H bond cleavage in the presence of $O_2$, the radical initialization of the B-based materials requires a high temperature to proceed, which decreases the thermodynamic advantages of the oxidative dehydrogenation reaction. Here, we report that the boron oxide overlayer formed in situ over metallic Ni nanoparticles exhibits extraordinarily low-temperature activity and selectivity for the ODHP reaction. With the assistance of subsurface Ni, the surface specific activity of the $BO_x$ overlayer reaches 93 times higher than that of bare boron nitride. A mechanistic study reveals that the strong affinity of the subsurface Ni to the oxygen atoms reduces the barrier of radical initiation and thereby balances the rates of the BO·H cleavage and the regeneration of boron hydroxyl groups, accounting for the excellent low-temperature performance of Ni@$BO_x$/BN catalysts.

Propene is one of the most important chemical feedstocks and building blocks of polymers[1–3]. The oxidative dehydrogenation of propane (ODHP) is a promising pathway to synthesize propene, as it is an exothermic reaction with no thermodynamic limitation and prevents the frequent carbon deposition removal and catalyst regeneration that occur during the commercialized propane direct dehydrogenation process (PDH)[4–9]. Although the ODHP process is a low-temperature favorable reaction that is more energy-efficient in theory, it has not yet been implemented on a large scale. The lack of highly selective catalysts to effectively prevent the overoxidation of propane into carbon oxides is one of the most critical challenges.

Boron-based nonmetal materials, including hexagonal boron nitride[10–14], supported boron oxide[15–18] and boron-doped zeolites[19–22], are widely reported to be selective to olefins and to suppress the

[1]Key Laboratory of Biomass Chemical Engineering of Ministry of Education, College of Chemical and Biological Engineering, Zhejiang University, 310027 Hangzhou, China. [2]Beijing National Laboratory for Molecular Sciences, College of Chemistry and Molecular Engineering and College of Engineering, Peking University, 100871 Beijing, China. [3]State Key Laboratory of Coal Conversion, Institute of Coal Chemistry, Chinese Academy of Sciences, Post Office Box 165, 030001 Taiyuan, Shanxi, China. [4]Department of Chemistry, Guangdong Provincial Key Laboratory of Catalysis, Southern University of Science and Technology, 518055 Shenzhen, China. [5]Institute of Industrial Catalysis, State Key Laboratory of Green Chemistry Synthesis Technology, College of Chemical Engineering, Zhejiang University of Technology, 310014 Hangzhou, Zhejiang, China. [6]These authors contributed equally: Xiaofeng Gao, Ling Zhu, Feng Yang. ✉e-mail: wxd@sxicc.ac.cn; dma@pku.edu.cn; yaosiyu@zju.edu.cn

production of undesirable overoxidation byproducts. The unique behaviors of oxygen activation into active surface and gas-phase radicals rather than lattice oxygen over traditional metal or metal oxide systems have been proposed as the principal reasons for the remarkable performance of boron-based catalysts[10,12,23,24] because the radicals are highly selective for H-abstraction from hydrocarbon molecules and are difficult to insert into C–H and C–C bonds. Despite the excellent selectivity control, the initialization of the radicals on the boron-based materials is difficult at low temperature due to the high barrier of the homolytic dissociation of the O−H bond of the boron hydroxyl in the presence of dioxygen[25,26]. As a result, the B-based catalysts must function above 500 °C to obtain sufficient activity, which decreases the thermodynamic advantages of the ODHP process. Simultaneously, the high working temperature also aggravates the mobility of the hydroxylated boron centers, leading to quick deactivation of the catalysts[27–29]. Therefore, enhancing the low-temperature ODHP activity of boron-based materials is necessary but highly challenging for application. Weakening the O−H bond of the boron hydroxyl, reducing the barrier for $O_2$-assisted O−H bond cleavage and accelerating the dissociation-regeneration cycle between BO• and BO-H (BO · ↔ BO·H) species are critical factors that need to be achieved.

Here, we report that the boron oxide shell formed in situ over metallic Ni nanoparticles exhibits extraordinary low-temperature activity and selectivity for the ODHP reaction. Over 25% conversion of propane can be achieved over the Ni@$BO_x$/BN catalyst at only 440 °C with better propane selectivity than that of conventional nonmetal boron-based catalysts. Under the influence of the Ni core, the surface specific propene formation rate of the $BO_x$ shell is 8 $\mu mol_{C3H8}/(m^{2*}s)$, nearly 2 orders of magnitude greater than that of unpromoted boron nitride. Mechanistic studies reveal that the mismatch of the Ni surface and the boron oxide lattice elongates the B−O bond and thus weakens the O−H bond of the boron oxide overlayer. The interaction between the subsurface Ni and the oxygen atom facilitates the cleavage of the O−H bond of the boron hydroxyl intermediate, reducing the energy barrier of the $O_2$-assisted O−H bond dissociation to 0.26 eV, which significantly enhances the radical initialization efficiency at low temperature. The discovery of the promotion effect of the subsurface metal centers on the surface of boron oxide provides a new route to construct highly active, selective and stable catalysts for the low-temperature oxidative dehydrogenation of propane.

## Results and discussion

### Ni@$BO_x$/BN structure and characterization

Ni-boron nitride hybrid materials were prepared via the successive synthesis procedures illustrated in Fig. 1a. The supported Ni/h-BN (hexagonal boron nitride nanosheet as support, Fig. 1b) was prepared using the wet impregnation method. As shown in Fig. 1c, the Ni particles were evenly dispersed on the support with a relatively uniform size (4% loading, approximately 10 nm, Figs. S1 and S2). After treatment in a diluted $CO_2$ flow at 800 °C, the h-BN substrate was etched by the Ni particles. Driven by the so-called reactive metal support interaction, the leached boron species encapsulated the Ni NPs and formed an amorphous boron oxide overlayer (denoted as Ni@$BO_x$/BN, Fig. 1d; S1 and S2). The electron microscopy images confirmed that the average shell thickness of the core-shell structure was 0.5-2 nm and the average diameter of Ni was around 25 nm. Electron energy loss spectroscopy (EELS) element mapping of a representative Ni particle (Fig. 1g; S3 and S4) showed that B and O coexisted in the shell. The distances between the diffraction fringes of the shell and core were found to be 2.09 and 2.03 Å, respectively (Fig. 1f and S5), identical to the $B_2O_3$ (111) and Ni (111) crystal planes. Therefore, it was confirmed that the core-shell structure observed was composed of a metallic Ni(0) core and a boron oxide shell. Furthermore, the boron oxide overlayer on Ni NPs was selectively removed by soaking Ni@$BO_x$/BN in boiling water (Fig. 1h,

Ni/BN-W). After washing with diluted nitric acid, all the exposed Ni NPs in the Ni/BN-W were leached out (Fig. 1i, Ni/BN-A), leaving only the porous BN substrate. XRD (Fig. S6), $N_2$ adsorption–desorption (Table S1) and elemental analysis (Table S2) results also revealed that the Ni was in a metallic state, and the texture parameters of BN did not change after the synthesis processes. To understand the structure of the as-prepared catalysts and to further confirm whether the metallic Ni core was fully encapsulated by the $BO_x$ overlayer, detailed characterizations were carried out. The in situ powder X-ray diffraction (XRD) patterns of Ni/BN and Ni@$BO_x$/BN both showed the diffraction signals of face-centered cubic structured Ni after reduction. However, after being treated in $O_2$ at elevated temperatures, the supported Ni particles of Ni/BN converted to NiO at only 300 °C. In contrast, the $BO_x$-encapsulated Ni NPs remained metallic without oxidation even after treatment at 600 °C, indicating that the $BO_x$ overlayer was intact and fully encapsulated the Ni(0) core (Fig. 1j and S7). Therefore, the $BO_x$ shell was able to protect the Ni(0) NPs under ODHP reaction conditions. The complete encapsulation of the prepared Ni@$BO_x$/BN catalyst was also confirmed by $H_2$ chemisorption (adsorption capacity <0.0001 ml $g^{-1}$, Table S3) and CO probe IR spectroscopy (Fig. S8). In contrast, a large $H_2$ adsorption capacity (0.30306 ml $g^{-1}$) and intense signals of linear and bridged-coordinated CO on Ni (2033 and 1903 $cm^{-1}$) were observed for both the 4%Ni/BN and Ni/BN-W catalysts, suggesting that CO and $H_2$ cannot penetrate the $BO_x$ shell. The in situ XPS spectra suggested that the Ni particles were metallic after reduction and $CO_2$ treatment (852.6 eV in the Ni 2$p$ region, Fig. 1k). $CO_2$ treatment at 800 °C increased the surface oxygen species in the catalysts (192.2 eV signal in the B 1$s$ region, attributed to $B_2O_3$, see Fig. 1k and Fig. S9).

### Low-temperature ODHP performance of Ni@$BO_x$/BN

The low-temperature catalytic performance of Ni@$BO_x$/BN, BN and three other nickel-BN hybrid materials was evaluated in the ODHP reaction at 440 °C (Fig. 2a). The 4% Ni@$BO_x$/BN catalyst achieved over 25% $C_3H_8$ conversion, with selectivities of 68% and 80% for $C_3H_6$ and total olefins (selectivity of $CO_2$ below 3%, Fig. 2a and S10). In comparison, h-BN only converted ~7% of propane under the same conditions, similar to previous reports[16,30]. Poor activity was also observed on porous BN (Ni/BN-A, $C_3H_8$ conversion less than 2%). These results confirmed that neither BN nor porous BN could function at relatively low temperatures. The presence of Ni NPs was necessary for the significantly enhanced low-temperature activity of the boron oxide overlayer. On the other hand, nearly 100% propane conversion was observed over the reduced 4% Ni/BN supported catalyst. However, the products of the reaction consisted of 60% methane and ~40% carbon oxides. The large amount of undesirable products formed over exposed Ni NPs suggested that complete encapsulation of the transition metal center is also necessary to prevent C−C bond cleavage and overoxidation side reactions. Indeed, when the $BO_x$ overlayer was selectively removed from the Ni@$BO_x$ composite, the selectivity of $CO_2$ was restored to >70% over the Ni/BN-W catalyst. We also demonstrated that the NiO/BN composite was inert for the ODHP reaction (Fig. S11). Therefore, the comparative performance evaluation revealed that the $BO_x$ overlayer fully encapsulating the metallic Ni NPs was the active site for the low-temperature ODHP reaction in the 4% Ni@$BO_x$/BN catalysts. Compared with other nonmetal B-based ODHP catalysts, including B/$SiO_2$, B/BN, and B/MWW zeolite (Fig. 2b), the 4% Ni@$BO_x$/BN catalyst exhibited much better low-temperature activities in the temperature range from 400 to 480 °C. The conversion of propane over the Ni@$BO_x$/BN catalyst was almost double that of the other counterparts at 440 °C. To estimate the activity of the subsurface Ni-modified $BO_x$ overlayer in the ODHP reaction, we tuned the loading of Ni and prepared a series of Ni@$BO_x$/BN using the same procedure (Figs. S12 and S13). When the loading of Ni increased from 2 to 4%,

the conversion of $C_3H_8$ increased slightly (Fig. 2c). A further increase in the Ni loading resulted in a sharp drop in activity (from 24% to 8%), indicating that the promotion effect was independent of the Ni loading on the catalysts. Furthermore, the contribution of the $BO_x$ overlayer to the overall activity of Ni@$BO_x$/BN catalysts was calculated by excluding the activity of the BN support (Fig. 3d; Tables S4 and S5). A linear correlation was observed between the surface area of the $BO_x$ overlayer and the BN-free activity of the catalysts, strongly suggesting that the activities of the $BO_x$ overlayers were similar to each other. The surface normalized $C_3H_8$ conversion rate of the $BO_x$ overlayer was approximately 8.11 μmol/($m^2$*s), 93 times higher than that of the BN support. With these highly active surface $BO_x$ overlayers as the active sites, the 4% Ni@$BO_x$/BN catalyst showed remarkable propene space-time yield. At a high space velocity (56 L/($g_{cat}$*h)), 3.6 and 5.6 g/($g_{cat}$*h) olefin productivities were achieved at 26% and 47% propane conversions, much better than the nonmetal boron-based catalysts reported previously (Figs. S14 and S15). The influence of mass and heat transfer has also been evaluated and the results also confirmed that the promotion effect of subsurface Ni core is significant (Fig. S16). Additionally, the

conversion-selectivity profiles (Fig. 2e) demonstrated that the selectivity of the propene and total olefins of Ni@$BO_x$/BN were even better than those of h-BN, especially at high $C_3H_8$ conversion. The nanometer-thick boron oxide shell over the Ni NPs also showed good structural stability due to the reduced working temperature. In the 50-h stability test at 440 °C, the conversion of propane over Ni@$BO_x$/BN decreased only slightly from 23.4% to 22.1%. The STEM images of the spent catalyst suggested that the core-shell Ni@$BO_x$ structure was intact after the reaction (Fig. S17). Other spectroscopic characterizations also confirmed that the chemical properties of Ni@$BO_x$/BN remained unchanged, except that more $B_2O_3$ species were formed on the BN support after the ODHP reaction (Figs. S18 and S19; Table S6).

## The crucial effect of subsurface nickel on the catalytic behavior of overlayer $BO_x$ in ODHP reaction

To understand the catalytic behavior of the Ni@$BO_x$/BN catalysts, detailed mechanistic studies were performed to gain kinetic insights. The dependence of the reaction rate on the partial pressure of oxygen and propane was almost identical over Ni@$BO_x$/BN and BN (Fig. 3a, b).

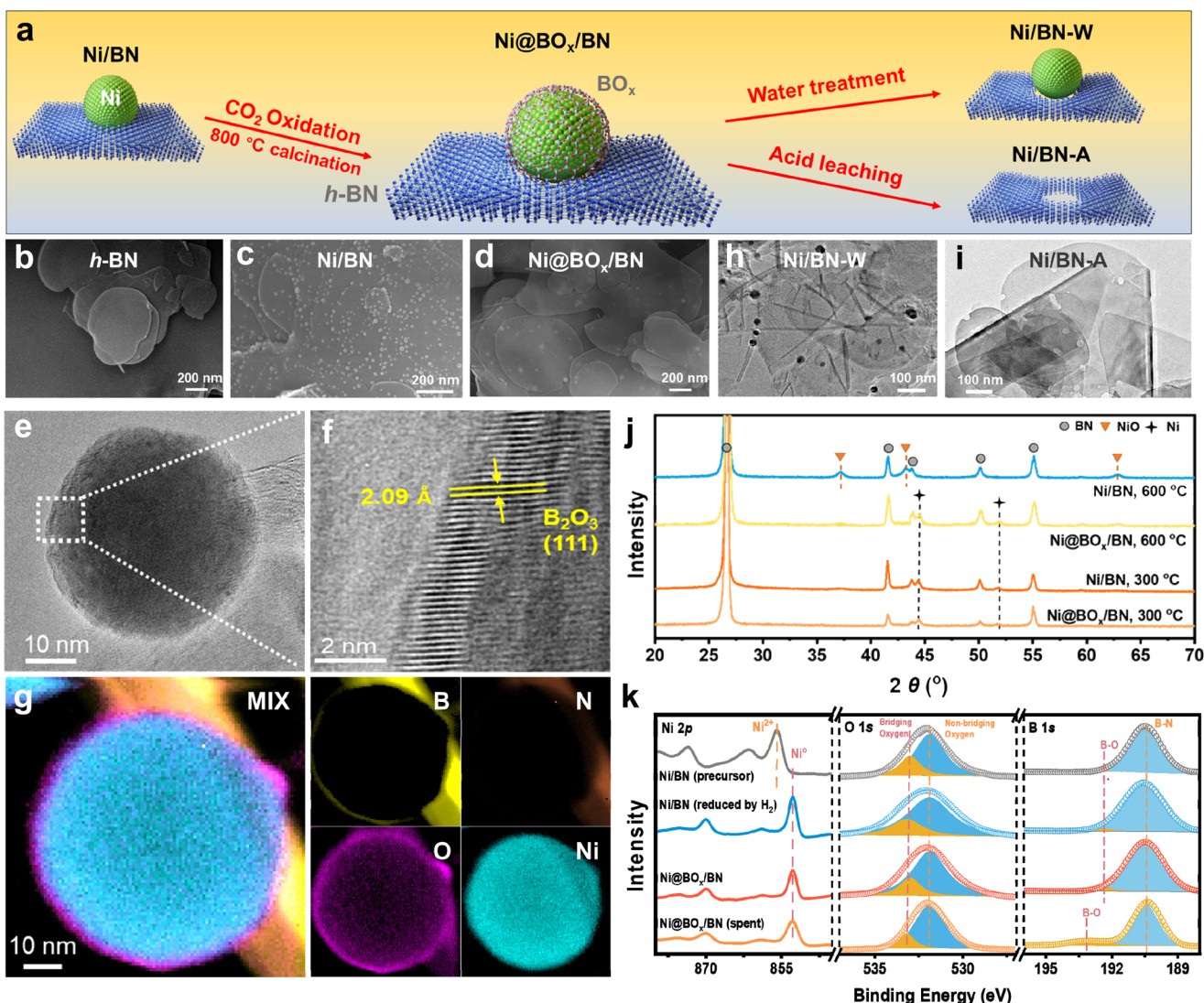

**Fig. 1 | Structural characterization of the Ni-BN composites. a** A schematic illustration of the synthesis procedure of Ni-BN composites. **b–d** electron microscopy images of h-BN, Ni/BN and Ni@$BO_x$/BN. **e, f** High-resolution TEM images of a representative Ni@$BO_x$ particle and the core-shell interface of Ni@$BO_x$/BN. **g** EELS element mapping of a representative Ni@$BO_x$ particle. **h, i** TEM images of Ni/BN-W and Ni/BN-A. **j** In situ XRD profiles of the Ni-BN composites under forced oxidation conditions. **k** In situ XPS characterization of the Ni-BN composites.

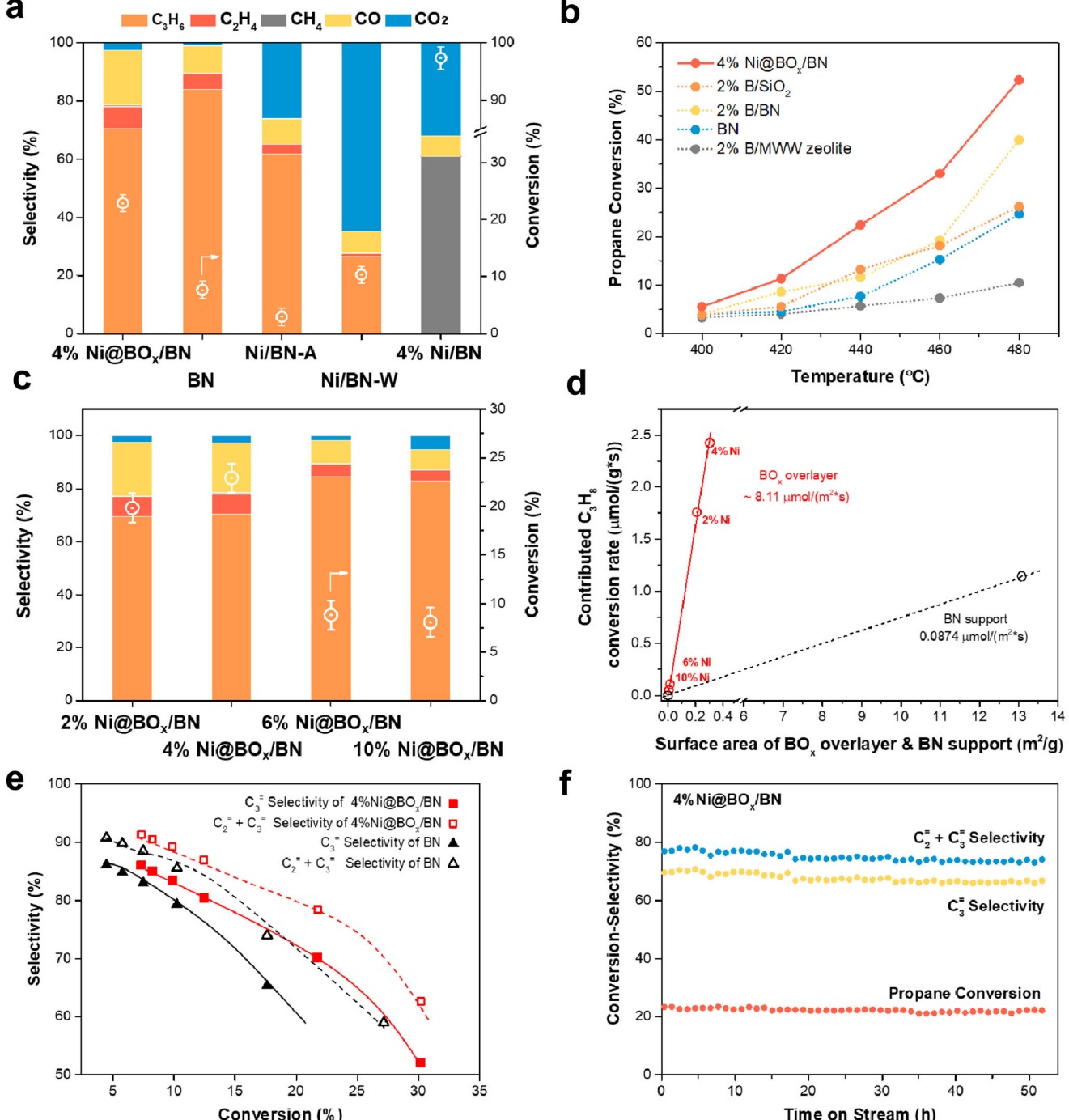

**Fig. 2 | Catalytic performance of Ni@BO$_x$/BN catalysts in the oxidative dehydrogenation reaction of propane (ODHP). a** Comparison of the activity and selectivity of the Ni@BO$_x$/BN catalyst with h-BN and other Ni-BN composites for the ODHP reaction. **b** Temperature-dependent performance of Ni@BO$_x$/BN and other nonmetal boron-based ODHP catalysts. **c** The catalytic performance of Ni@BO$_x$/BN catalysts with different Ni loadings. **d** The dependence of the propene formation rate on the surface area of the Ni core in Ni@BO$_x$/BN. **e** Conversion selectivity of the BN and Ni@BO$_x$/BN catalysts. **f** stability test of the Ni@BO$_x$/BN catalyst; typical conditions: 100 mg catalyst, 440 °C, C$_3$H$_8$/O$_2$/N$_2$ = 1:1.5:3.5, 0.1 MPa, WHSV = 7200 L·kg$^{-1}$·h$^{-1}$.

The Eley-Rideal-like kinetic behavior of O$_2$ and the second-order dependence of p(C$_3$H$_8$) on the reaction rates at constant p(O$_2$) suggested that the ODHP reaction over Ni@BO$_x$/BN was mainly controlled by the free radical intermediates rather than the surface reaction. Dilution of Ni@BO$_x$/BN with inert silicon carbide (SiC, Fig. 3c) induced an unusual improvement in the mass-specific activity (increased by 50% at a 1:1 dilution), which also demonstrated that the oxidative dehydrogenation of propane was not a surface-controlled reaction over the Ni@BO$_x$/BN catalyst. The increase in activity after dilution was

the result of a significant reduction in the chain termination probability of the active radicals upon the physically separated active sites[23,25,26]. Time-resolved in situ diffuse reflectance Fourier transform infrared spectroscopy (DRIFTS) was used to understand the promotion of the radical pathway by the encapsulated Ni NPs (Fig. 3d–f). In previous studies[10,17,25], the surface boron hydroxyl groups have been widely studied and proven to be the active sites for the ODHP reaction. In the reaction, the O–H bond of the boron hydroxyl dissociated homolytically, released the surface B-O· and gas phase ·OOH radicals and

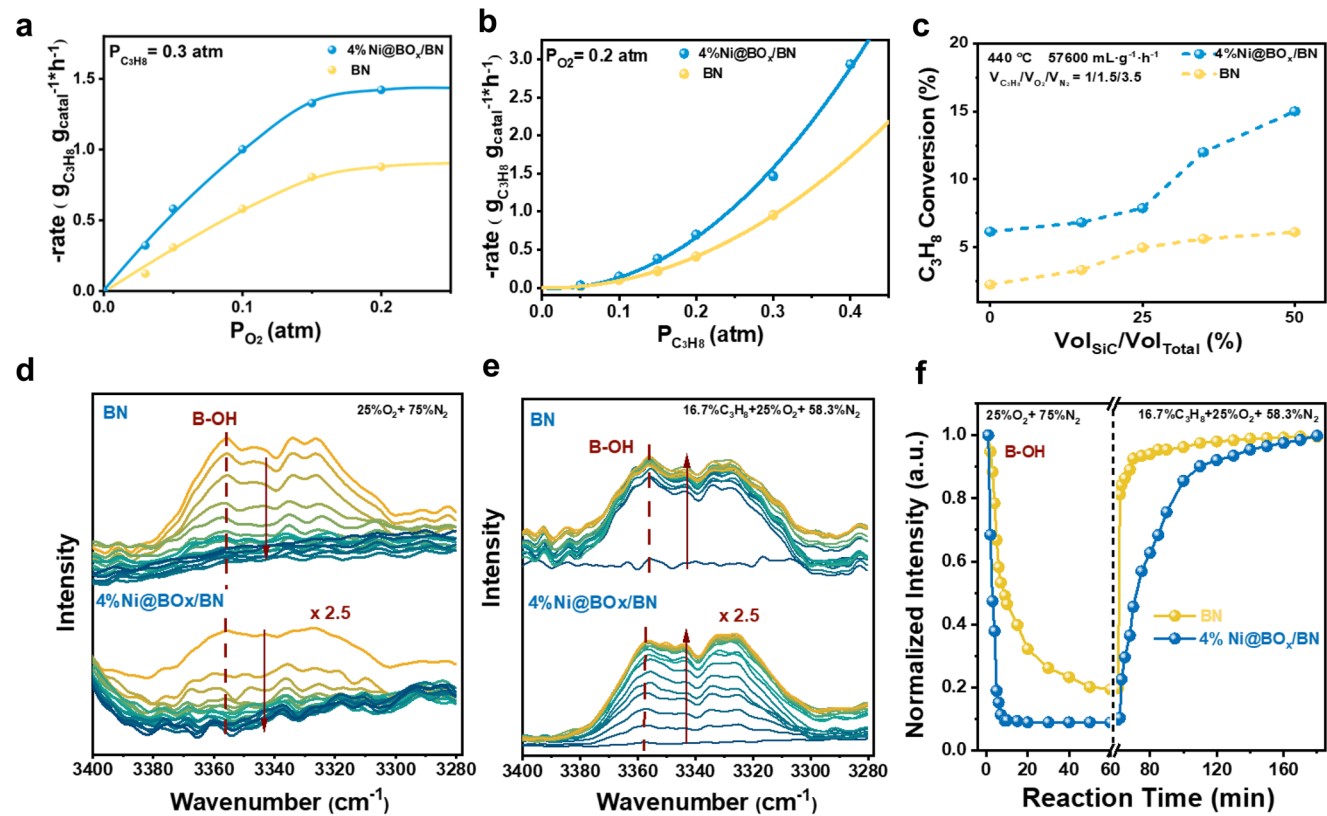

**Fig. 3 | Experimental studies of the reaction mechanism of the ODHP reaction over the Ni@BO$_x$/BN catalyst. a** Dependence of the reaction rate on p(O$_2$). **b** Dependence of the reaction rate on p(C$_3$H$_8$). **c** The change in reaction rate on the dilution ratio. **d**, **e** Time-resolved in situ DRIFTS observation of the B-OH species under transient consumption and formation processes. **f** Normalized intensity of the O–H stretching vibration peak with time under transient consumption and formation processes.

started the radical chain reaction (the initialization step of radicals, BO·H + O$_2$ = BO· + ·OOH(g)). Owing to the high bonding tendency, the B-O· and ·OOH radicals abstracted hydrogen directly or indirectly from propane molecules with a low barrier and generated ·C$_3$H$_7$ radicals, which tended to dehydrogenate into propene with high selectivity[25-27]. Simultaneously, the boron hydroxyl groups were regenerated and ready for the next cycle of radical initialization. Based on the mechanistic understanding, we designed a two-step transient DRIFTS experiment recording the O–H stretching vibration band of the boron hydroxyl (3360–3320 cm$^{-1}$) to monitor the radical initialization and regeneration activities of the active boron hydroxyl species. Before the observation, the Ni@BO$_x$/BN was exposed to the reaction feed (C$_3$H$_8$ and O$_2$ mixture 1:1.5) to reach a steady state (Fig. S20). The propane in the gas feed was then switched off and replaced with N$_2$ with the same flow rate used in the first step to observe the depletion of the hydroxyl group in O$_2$. After all the O–H groups were consumed, the sample was further exposed to the reaction atmosphere to monitor the regeneration of the hydroxyl species (Fig. 3d, e). When C$_3$H$_8$ was removed from the gas feed (left panel, Fig. 3f), the consumption rate of the surface hydroxyl on Ni@BO$_x$/BN was faster than that of the BN support, indicating that radical initialization was accelerated. In other words, the generation rates of both BO· and ·OOH radicals were higher on the Ni-modified BO$_x$ overlayer than on the unpromoted BN supports. When propane was reintroduced into the gas feed, the O–H band reappeared on both BN and Ni@BO$_x$/BN catalysts, indicating the regeneration of the boron hydroxyl species. The rate of O–H regeneration over the BO$_x$ overlayer was much lower than that over h-BN (it took 100 min to reach 90%; see the right panel of Fig. 3f). The slow accumulation of boron hydroxyl species under the reaction conditions suggested that the rate of O–H bond formation was similar to that of

dissociative bond depletion, resulting in a slow net increase in boron hydroxyl in the reaction gas feed. In contrast, the rapid saturation of the boron hydroxyl species on the BN sample indicated that the hydrogen abstraction activity of the BO· in the hydroxylated edge site of the BN support far exceeded that of the O–H cleavage. As a result, it was anticipated that the BO$_x$ overlayer could release radicals at a higher turnover frequency than hydroxyl-functionalized BN because the active site of BN tended to be occupied by H atoms[25,31]. From these phenomena, it was confirmed that the main role of the encapsulated Ni NPs in the ODHP reaction is to promote the dissociative activation of boron hydroxyl and accelerate the initialization of the surface and gas-phase free radicals.

To obtain mechanistic insight on the function of the encapsulated Ni NPs, the structure of Ni@BO$_x$/BN and the reaction pathway for the ODHP reaction were investigated using density functional theory (DFT) calculations. The fully encapsulated Ni@BO$_x$ active site was constructed by placing one layer of B$_2$O$_3$ on the Ni(111) slab. After relaxation and structural optimization, an atomic B$_2$O$_3$ layer was formed, in which the oxygen atoms were bonded to the top site of the Ni surface and boron atoms occupied the threefold vacancies of the Ni surface (Fig. S21). The length of the B–O bond in the B$_2$O$_3$-Ni(111) model was 1.45 Å, 0.08 Å longer than that of the bulk B$_2$O$_3$ (1.37 Å). Bader charge analysis showed a significant increase in the electron density between the B$_2$O$_3$ overlayer and Ni(111) (Table S7 and Fig. S22). Owing to the change in the geometry and electronic properties, the electron loss of the H atom in the boron hydroxyl of the B$_2$O$_3$-Ni(111) model was reduced by 0.1 unit compared with that of bulk B$_2$O$_3$ (Table S7). That is, the O–H bonds were activated by the subsurface Ni. The reaction pathway was calculated on the optimized B$_2$O$_3$/Ni (111) model. Before the catalytic cycle of the ODHP

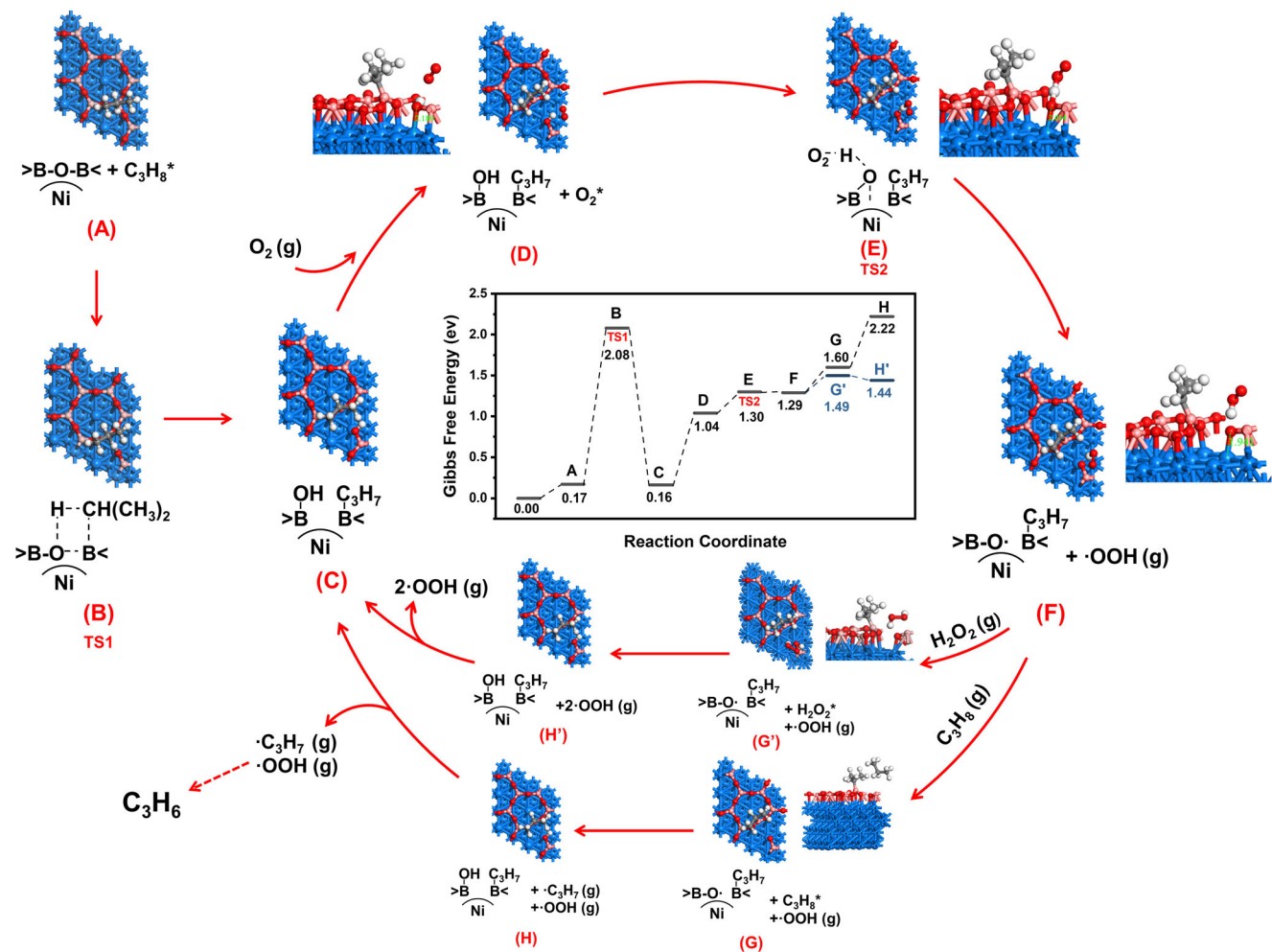

**Fig. 4 | DFT calculation of the ODHP reaction pathway on the $B_2O_3$/Ni(111) model surfaces.** The activation of the active site is shown from state A to state C. States C to H (H') illustrate the catalytic cycles of the ODHP reactions. The light blue, red, pink, gray and white atoms represent the Ni, O, B, C, and H atoms of the system. The Gibbs free-energy change of each step is presented in the figure of the reaction profile.

reaction (Fig. 4 and S23), a preactivation step was required to dissociate the B−O bond and form the boron hydroxyl active site (states A to C, $\Delta G^{\neq}$ (activation free-energy change) was 1.91 eV, with $\Delta G = -0.01$ eV). After activation, the boron hydroxyl group reacted with the adsorbed $O_2$ to initialize the radical reaction (states C to F). The effective activation free energy of this process was 1.14 eV, and $\Delta G = 1.13$ eV, so the reaction proceeded relatively easily at temperatures above 400 °C. Most importantly, the barrier of the $O_2$-assisted O−H cleavage (state D to F) was only 0.26 eV, much smaller than that of the same step on the nonmetal B-based materials[25,26]. The configurations of the corresponding states (Fig. S24) showed that the length of the O−Ni bond decreased from 2.18 to 1.94 Å (states D to F), indicating that the affinity of the subsurface Ni to the oxygen atoms in the $BO_x$ overlayer assisted the O−H bond dissociation in the transition state (Fig. S24). After radical initialization, the BO· radicals stabilized on subsurface Ni (111) participated in the catalytic cycle by abstracting a hydrogen atom from reactant $C_3H_8$ or intermediate $H_2O_2$ and regenerated the boron hydroxyl species (states F to C)[25,26]. The direct reaction of the BO· radical with $C_3H_8$ is a surface conversion pathway of propane (F → G → H → C, the full reaction network was presented in Table S8a). While the H abstraction from $H_2O_2$ can be regarded as a gas-phase conversion pathway of propane (F → G' → H' → C, the full reaction network was presented in Table S8b). The free-energy change of the BO· radical assisted H abstract from adsorbed propane (F to H) was 0.93 eV,

similar to the $\Delta G^{\neq}$ of the radical initiation step (C to F). The thermodynamic barrier of the H-abstraction step (G to H) is only 0.62 eV. Moreover, the free-energy change of H abstract from $H_2O_2$ intermediate is even lower (F to H', 0.15 eV) with a thermodynamic barrier of 0.20 eV, indicating the gas-phase $C_3H_8$ conversion pathway is thermodynamically more favorable, which agrees with previous studies[25,26]. Nevertheless, the turnover frequency of the regeneration of the BO-H active site and O−H cleavage are similar based on the calculation results, consistent with the DRIFTS observation. In comparison, the BO-H homolytic cleavage barrier over nonmetal boron-based materials are generally high (above 2.0 eV) based on the previous theoretical studies[25,26]. As a result, the low barrier of $O_2$-assisted O−H dissociation and the similar energy requirements for the generation of active oxygen radicals and BO-H regeneration steps ensured an efficient low-temperature radical initialization of the Ni@$BO_x$/BN catalyst, which was the major reasons for the excellent performance of in the low-temperature ODHP reaction.

In summary, our discovery demonstrates that encapsulated metallic Ni nanoparticles significantly enhance the reactivity of the boron oxide overlayer, providing an opportunity to reduce the working temperature of boron-based catalysts during the oxidative dehydrogenation of propane. The subsurface Ni induces B−O bond extension in the $BO_x$ overlayer and electron transfer from Ni to the $BO_x$ layer. The appropriate affinity of subsurface Ni for the oxygen atoms in

the $BO_x$ overlayer lowers the barrier of the radical initialization step and thereby balances the rate of BO-H cleavage and boron hydroxyl regeneration, which accounts for the excellent low-temperature performance of the Ni@$BO_x$/BN catalyst. The discovery of the promoting effect of metal NPs on boron-based materials presents a step forward in the development of active, selective and robust low-temperature catalysts that can fully exploit the thermodynamic advantages of the ODHP reaction.

## Methods

### Materials
Hexagonal boron nitride (BN) was purchased from Alfa Aisa Inc. Nickel nitrate hexahydrate (Ni($NO_3$)$_2$·$6H_2O$) and ethanol ($C_2H_5OH$) were purchased from Shanghai Aladdin Bio-Chem Technology Co., Ltd. These raw materials were used as received without further treatment.

### Synthesis
x%Ni/BN: x%Ni/BN was prepared by an impregnation method. In a typical synthesis process, firstly, the hexagonal boron nitride (BN) was dispersed into ethanol with the amount ratio of $m_{BN}$: $V_{ethanol}$ = 1 g:10 ml, and the suspension was sonicated for 30 min. Then, the required amount of nickel nitrate hexahydrate (Ni($NO_3$)$_2$·$6H_2O$) was added to the suspension by controlling the nickel loading to x% wt. The resulting mixed solution was evaporated to powder at room temperature under strong mechanical agitation, and subsequently placed in an oven to dry overnight at 70 °C. Finally, the samples were further treated in a pure hydrogen stream at 500 °C for 2 h to obtain the x%Ni/ BN catalysts, where the x was denoted as mass fraction of Ni element, and 2 wt%, 4 wt%, 6 wt% and 10 wt% were employed.

The Ni@$BO_x$/BN catalysts were obtained by the oxidation of x%Ni/ BN samples in a 20 vol% $CO_2$/$N_2$ mixed gas flow at 800 °C for 2 h.

The Ni/BN-W catalyst was obtained by immersing Ni@$BO_x$/BN catalyst into boiled water under a strong mechanical agitation for 2 h, drying overnight at 70 °C and heating in a pure hydrogen stream at 500 °C for 2 h.

The Ni/BN-A catalyst was obtained by treating Ni@$BO_x$/BN catalyst with 10% $HNO_3$ under a strong mechanical agitation for 12 h, drying overnight at 70 °C and heating in a pure hydrogen stream at 500 °C for 2 h.

### Catalyst characterization
The (in situ) powder diffraction patterns (XRD) of the Ni/BN series catalysts were collected using an XPERT-3 diffractometer with Cu Kα1 radiation (1.540598 Å) as the incident X-ray. The scan rate was set at 2°/min with a step of 0.02°. The mass-specific surface areas were determined using $N_2$ physisorption method. The experiments were performed using a BELSORP-mini instrument. The loadings of Ni in the Ni/BN series catalysts were determined using the inductive coupling plasma atomic emission spectroscopy (ICP-AES) over a Varian ICP-OES-720 instrument. In the sample preparation, the Ni portion was dissolved from catalyst powder by aqua regia. The resulted solution was then diluted to desirable concentration for further measurement. The STEM images of the Ni/BN series catalysts were collected by a JEOL 2100 microscope equipped with EELS analyzer. The Ni/BN series catalysts were dispersed in ethanol and sonicated to make the suspension of catalysts. One drop of the suspension was transferred to the carbon grid as the sample for electron microscopy characterization. The scanning electron microscopy (SEM) investigation was carried out with a Hitachi FESEM SU8220 electron microscopes. The in situ DRIFTS of Ni/BN series catalysts for propane oxidative dehydrogenation under reaction conditions were recorded using a Nicolet iS 20 FT-IR spectrometers equipped with an MCT-A detector, KBr windows and a HARRICK in situ cell. 0.1 g of the sample was loaded into the in situ

reaction cell and was in situ pretreated in a flow (20 ml min$^{-1}$) of $N_2$ at 200 °C for 1 h and then heated to reaction temperature (440 °C). The background spectrum was collected in the continuous $N_2$ flowing. A mixture gas of 16.7 vol% $C_3H_8$ + 25 vol% $O_2$ + 58.3 vol% $N_2$ (5 ml min$^{-1}$) was subsequently introduced into the reaction cell, and the spectra were collected for 4 h. After that, the mixture gas was instantaneously switched to 25 vol% $O_2$ + 75 vol% $N_2$ (5 ml min$^{-1}$) and the spectra were collected for another 2 h. Finally, the mixture gas was instantaneously switched back to 16.7 vol% $C_3H_8$ + 25 vol% $O_2$ + 58.3 vol% $N_2$ (5 ml min$^{-1}$) and the spectra were collected for 3 h. The DRIFT spectra were obtained in the range of 4000 to 650 cm$^{-1}$ with a resolution of 4 cm$^{-1}$ and 64 scans. The CO probe DRIFTS of the 4% Ni/BN, 4% Ni@$BO_x$/BN and Ni/BN-W catalysts were collected using a Bruker Vortex 80 spectrometer equipped with diffused reflectance accessories. The fine powders of the catalysts were loaded in the cell and the background spectrum was collected at room temperature in Ar flow. Then, the sample was exposed to 20% CO in Ar until reaching steady state. The cell was purged with Ar flow to remove physiosorbed CO and the IR spectra were collected in the meantime until no change occurred. The last spectrum was used for analysis. The (In situ) X-ray photoelectron spectra (XPS) analysis was performed using a Thermo Scienti fic ESCALAB 250Xi spectrometer using an Al K α X-ray source and pass energy of 20 eV. The C 1 s peak located at 284.5 eV was used to calibrate binding energy positions. $H_2$ Pulse chemisorption was recorded using an AutoChemII 2920 station from Micromeritics. 100 mg x% Ni/BN-W catalysts were loaded into U-shaped quartz reactor with an inner diameter of 0.5 cm. Before the test, the sample was pretreated in a flow (30 ml min$^{-1}$) of $H_2$ at 500 °C for 1 h and then maintained 400 °C for another 30 min with switching to an argon flow of 30 ml min$^{-1}$. After that, the sample was cooled to 30 °C for chemisorption in order to clean the Ni surface and to avoid the presence of residual adsorbed hydrogen. The $H_2$ pulse chemisorption was performed at 30 °C and the volume of the injection loop was 0.5 cm$^3$. The carrier gas was Ar in the case of $H_2$ pulses. The $H_2$ consumptions were measured by a thermal conductive detector (TCD).

### Catalytic performance test
Taking 4%Ni@$BO_x$/BN catalyst as an example, in a typical experiment, 200 mg of 4%Ni@$BO_x$/BN catalyst was loaded in a fixed bed tube furnace reactor with an inner diameter of 8 mm. The reactant gas consists of 16.7 vol% $C_3H_8$, 25 vol% $O_2$ and 58.3 vol% $N_2$, and a reactant flow rate of 24 ml min$^{-1}$ was employed with a weight hourly space velocity (WHSV) of 7200 ml/($g_{cat.}$ · h). The sample was heated to the designated reaction temperature (400–500 °C) in the flow of reactant gas. The products of the reaction were analyzed by an on-line gas chromatography (GC, Agilent 8890) equipped with a thermal conductive detector (TCD) and a flammable ionization detector (FID). The $N_2$ in the flow was used as the inner standard. The response factors of reactants and products were calibrated using standard curve methods.

### DFT calculations
The adsorption energy calculations were performed with the plane wave based pseudo-potential code in Vienna ab initio simulation package (VASP). The electron-ion interaction is described with the projector augmented wave method (PAW). The exchange and correction energies are described by the generalized gradient approximation using Perdew–Burke–Ernzerhof formulation (GGA-PBE). Since van der Walls dispersion interaction plays an important role in weak interaction system, the D3 correction of Grimme was used. The plane wave cutoff energy was specified by 400 eV, the electron smearing method with $\sigma$ = 0.20 eV was used to ensure energies with errors due to smearing of less than 1 meV per unit cell. The convergence criteria for geometry optimizations of total energy and forces were $10^{-5}$ eV and

0.02 eV Å⁻¹. Spin polarization was included. The $4 \times 4 \times 1$ Monkhorst-Pack k-points sampling was used. To avoid interactions among slabs, the vacuum layer between periodically repeated slabs was set as 15 Å. Transition state was located using the climbing image nudged elastic band (CI-NEB) method.

## Data availability

The data that support the plots within this paper and other finding of this study are available from the corresponding author upon reasonable request.

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

## Acknowledgements

S.Y. acknowledges the financial support from the Zhejiang Provincial Natural Science Foundation of China under Grant LR21B030001, Natural Science Foundation of China (22178302) and National Key R&D Program of China (2022YFB4003100, 2021YFC2101800) and Beijing National Laboratory for Molecular Sciences (BNLMS202003). L.L. acknowledges the Natural Science Foundation of China (22002140), Young Elite Scientist Sponsorship Program by CAST (2019QNRC001) and Zhejiang Provincial Natural Science Foundation of China (LR22B030003).

## Author contributions

S.Y., D.M., and L.L. designed the research. X.G. performed most of the reactions. Ling Z. and X.W. completed the theoretical calculations. F.Y. and Lei Z. performed the electron microscopy analyses. S.Y. and X.G. completed the paper. W.X., X.Z., Y.H., and H.S. performed some of the experiments and catalyst characterizations. All authors discussed the results and commented on the manuscript.

## Competing interests

The authors declare no competing interests.
