## [Peer Review File · Nature Communications]

Subsurface Nickel Boosts the Low-Temperature Performance of a Boron Oxide Overlayer in Propane Oxidative DehydrogenationREVIEWER COMMENTS

Reviewer #1 (Remarks to the Author):

The authors have studied Ni/BN catalysts and observed higher activity of these catalysts at comparatively low temperatures. It is proposed that encapsulated metallic Ni nanoparticles increase the reactivity of a boron oxide top layer. The interaction of Ni with oxygen atoms in the BO_x top layer is assumed to lower the barrier of the radical initialization step and thus accelerate BO-H bond cleavage. The catalyst shows higher conversion, but the selectivity is similar to Ni-free systems at comparable conversion.

However, neither the catalysts under working conditions nor the spent catalysts are comprehensively characterized in the paper. I suspect that under the oxidative and hydrothermal conditions of the reaction, the BN shell partially opens and nanostructured NiO is formed. NiO is an excellent low temperature catalyst for propane oxidation, see e.g. <https://link.springer.com/article/10.1007/s11244-020-01329-5>.

The formation of NiO must therefore be convincingly excluded, at least in the spent catalyst. In Fig. S17, for example, only the B 2p XPS spectra are shown, but not Ni spectra.

Reviewer #2 (Remarks to the Author):

Gao et al. report an active nickel-boron oxide catalyst for propane oxidative dehydrogenation. This particular catalyst can promote the reaction at temperatures relatively lower than regularly expected for conventional boron/boron nitride based catalysts with reasonably high selectivity propene. The authors show encapsulation of Ni with a BO_x overlayer yields the best performance, which is also modelled in the DFT calculations. However certain questions remain for the DFT portion of the work which are not convincing to explain the role of Ni in promoting the reaction. Overall these could be interesting results and are reasonably well-presented which are worth publishing in Nature Communications pending revisions

Specific comments below:

-Experimental thickness of boron oxide shell is 0.5-2 nm, corresponding to at least 2 atomic layers. How robust are the DFT conclusions for a two atomic layer model? For one layer, there is direct bonding between B and O with Ni, but this cannot be expected to be the case in the observed catalysts.

-In Fig 4, it would be helpful to mark transition states in the energy profile.

-I do not understand step G to H in the proposed mechanism: C-H abstraction is a barrierless process here, or have they simply not been calculated? Also, where is the second propane (C₃H₈^{*}) on the surface? There is only one C₃H₇ shown in the inset image for state G.

-Based on this mechanism, the rate limiting step is G to H, with a thermodynamic barrier of 1.45 and probably an even larger kinetic one. This contradicts with the conclusion that the reason for higher activity is due to the regeneration of boron hydroxyls.

-How stable is the propyl spectator in the surface? Is this verified from in situ spectroscopy? What is the barrier of the second C-H activation to form propene on the surface? This mechanism would not hold if it is easier for propyl to just continue to react and leave as propene, and the surface B-O* radical would consequently no longer be present.

Reviewer #3 (Remarks to the Author):

This contribution investigates the catalytic performance of boron oxide formed over metallic Ni nanoparticles in the oxidative dehydrogenation of propane (ODHP) by combined using of experimental and theoretical approaches. The authors successfully synthesize Ni@BOx/BN by using Ni-boron nitride hybrid materials after treatment at 800 °C in a CO₂ flow and this catalyst shows a remarkable low-temperature activity for ODHP compared to the conventional boron-based catalyst. In my view, this study provides another route for preparing a boron-based catalyst with improved activity and the catalytic properties have been well characterized. I have however some serious concerns as outlined below:

1. Although the Ni species encapsulated by BOx play a key role in the Ni@BOx/BN catalyst for ODHP, unlike the case of the Ni species, there is a lack of elucidation for understanding and supporting the role of B (e.g., BOx) species in the catalyst. Indeed, the catalysts with and without BOx species (i.e., Ni/BN-W and Ni@BOx/BN) exhibit the very different catalytic behaviors for ODHP. Thus, the details of the physicochemical properties of BOx in Ni@BOx/BN with the experimental data should be described more precisely.
2. As mentioned in this manuscript, the Ni@BOx/BN catalyst treated at 800 °C in a CO₂ flow shows increased surface oxygen species. If the surface oxygen species formed by CO₂ is attributed to BOx, which influences the catalyst activity, it is recommended to compare the catalyst properties treated under CO₂ and O₂, respectively. This will help to better understand the effect of BOx species on this reaction, and whether it can be controlled during the catalyst preparation step.
3. As shown in Figures 2 and 3, the Ni@BOx/BN catalyst shows the better performance than the conventional boron-based catalyst. However, the catalyst performance in ODHP can be over- or underestimated due to the heat and mass transfer limitations (Org. Process Res. Dev. 2018, 22, 1644–1652). Therefore, more in-depth discussions of the catalytic results are required to clearly explain the transfer limitation in the Ni@BOx/BN catalyst system in comparison with the conventional one.
4. Figure 4 summarizes the computational results. Step A => B => C is considered an initiation step, generating the active species in the catalytic cycle. This step is very questionable. Subsequently, the authors let adsorbed O₂ abstract an H-atom from BO-H to form both BO* and HOO* radicals. This step is problematic. The difference in BO-H and OO-H bond dissociation energy is likely much larger than the 0.25 eV (5.8 kcal/mol) predicted. The authors did not benchmark their computational level for such

open-shell species. Assuming O₂ would be able to abstract that H-atom, why would it not directly abstract H-atoms from propane as the C-H BDE in propane is weaker than X-O-H bonds? This part of the work is pretty shaky and needs to be addressed properly.

Response to the referees:

We thank the referees for the insightful and constructive comments and suggestions in order to help us further improve the manuscript. We have addressed all the comments point-by-point and revised the manuscript accordingly. In this response letter, comments from the referees are summarized in black typeface with the original comments quoted in *italic*, and our responses are in blue typeface. All major changes have been highlighted in blue in the main text. A detailed list of changes in the manuscript is also provided.

A list of main changes we have made:

1. We performed thorough investigation on influence of the heat/mass transfer and confirmed that the excellent performance of Ni@BO_x/BN is its intrinsic catalytic behavior.
2. The performance of NiO/BN was evaluated to exclude the contribution of nickel oxide on the activity of Ni@BO_x/BN catalyst.
3. The DFT calculation and data presentation was modified to explain the mechanism clearly.
4. The necessary data obtained during the revision was added into the manuscript and supporting information.

A point-to-point response to the comments

Referee #1 The reviewer 1 confirms the significance of our research. However, reviewer considers that the structure catalyst under working conditions and after reaction needs to be identified, especially whether exposed nickel oxide species are formed.

Comment 1. *“However, neither the catalysts under working conditions nor the spent catalysts are comprehensively characterized in the paper. I suspect that under the oxidative and hydrothermal conditions of the reaction, the BN shell partially opens and*

nanostructured NiO is formed. NiO is an excellent low temperature catalyst for propane oxidation, see e.g. <https://link.springer.com/article/10.1007/s11244-020-01329-5>.

The formation of NiO must therefore be convincingly excluded, at least in the spent catalyst. In Fig. S17, for example, only the B 2p XPS spectra are shown, but not Ni spectra..”

Reply: We appreciate the reviewer’s positive recommendation and valuable comments. We understand the concerns of the reviewer about the active site of the Ni@BO_x/BN catalysts. The core-shell structure of the Ni@BO_x on the BN support is robust under working conditions based on the following evidence.

1. The high-resolution STEM images and EELS element mapping of Ni@BO_x/BN-spent catalyst in Fig. S15 (see Figure R1) clearly show that the BO_x overlayer is intact after the ODHP reaction.

Figure R1. High resolution STEM images and EELS element mapping of Ni@BO_x/BN-spent catalyst

2. The Ni 2p region XPS spectra of the spent Ni@BO_x/BN catalyst was presented in Fig. 1k in the original paper. The Ni signals show typical metallic features and no NiO signal is observed, suggesting the Ni remain metallic under the protection of the BO_x overlayer in the propane-oxygen mixture. Additionally, we designed the forced oxidation experiment and monitored the valence of Ni core using in-situ XRD. The result also confirm that the Ni core is not oxidized until 700 °C in oxidative atmosphere.

Figure R2. In situ XPS characterization of the fresh and spent Ni/BN and Ni@BO_x/BN catalysts.

Meanwhile, the NiO is an excellent low temperature catalyst for propane oxidation. However, its propylene selectivity is very low in the ODHP reaction. From the recommended article (*Topics in Catalysis*, 63(19), 1731-1742.), the selectivity of propene of NiO is less than 20% under the C₃H₈ conversion of 5-20% (see Figure R3). While, under the same conversion, the selectivity of propene of the Ni@BO_x/BN catalyst is above 75%.

Figure R3. The conversion-selectivity profile to propene with the propane conversion of NiO-based catalysts (*Topics in Catalysis*, 63(19), 1731-1742)

Therefore, based on the characterization results and the catalytic performances, the nanostructured NiO species do not exist in Ni@BO_x/BN even under reaction conditions.

Referee #2

Gao et al. report an active nickel-boron oxide catalyst for propane oxidative dehydrogenation. This particular catalyst can promote the reaction at temperatures

relatively lower than regularly expected for conventional boron/boron nitride based catalysts with reasonably high selectivity propene. The authors show encapsulation of Ni with a BO_x overlayer yields the best performance, which is also modelled in the DFT calculations. However certain questions remain for the DFT portion of the work which are not convincing to explain the role of Ni in promoting the reaction. Overall these could be interesting results and are reasonably well-presented which are worth publishing in Nature Communications pending revisions

Comment 1. “Experimental thickness of boron oxide shell is 0.5-2 nm, corresponding to at least 2 atomic layers. How robust are the DFT conclusions for a two atomic layer model? For one layer, there is direct bonding between B and O with Ni, but this cannot be expected to be the case in the observed catalysts.”

Reply: It has been demonstrated the oxygen functionalization and formation of boron hydroxyl are common among boron based catalyst under the ODHP reaction condition.¹⁻³ Meanwhile, due to the low melting point of boric acid, the surface of the working catalyst is highly dynamic. Therefore, the balance of leaching and SMSI-driven encapsulation of the working Ni@BO_x catalyst will be built (Scheme R1). This phenomenon explains why the thickness of the core-shell catalyst is not identical. Meanwhile, it also confirms that the BO_x shell is loose enough to allow the reactant molecules to contact the inner shell of boron oxide. Therefore, all the BO_x species may participate the reaction, but only the first one- or two-layers BO_x shells are promoted by the subsurface Ni. In this sense, the simplification of Ni@BO_x/BN with single layer of BO_x covering the Ni NPs is appropriate, considering one of the main targets of this manuscript is to understand the promotion effect of encapsulated Ni NPs.

Scheme R1. The dynamic behavior of the BO_x overlayer under working condition

Actually, the modeling of the Ni@BO_x/BN catalyst with the single atomic layer of BO_x has been applied by Fu et al. to explain the catalytic performance of methane dry reforming reaction.⁴ Certainly, the simulation of the core-shell catalyst with multiple BO_x layers is important to evaluate the influence thickness of subsurface metal promoters. We will try to perform theoretical studies in the future studies.

Comment 2. *“In Fig 4, it would be helpful to mark transition states in the energy profile.”*

Reply: Thanks for your suggestions. We've calculated all the transition states of the reaction, and the energy profile of each transition states has been marked in the center of Fig. 4

Comment 3. *“I do not understand step G to H in the proposed mechanism: C-H abstraction is a barrierless process here, or have they simply not been calculated? Also, where is the second propane (C3H8*) on the surface? There is only one C3H7 shown in the inset image for state G.”*

Reply: The C-H abstraction (Step G to H) is the regeneration step of the surface boron hydroxyl. In our calculation, only the Gibbs free energy change was estimated. We are very sorry that we forget to add the adsorption step of propane and desorption steps of free radicals into the calculation. After the adsorption is included, the thermodynamic barrier of the C-H abstract and surface B-OH regeneration step is actually an endothermic process with the free energy change of 0.62 eV (see revised Fig.4). The presentation and discussion of the theoretical calculation was modified in the revised manuscript.

Comment 4. *“Based on this mechanism, the rate limiting step is G to H, with a thermodynamic barrier of 1.45 and probably an even larger kinetic one. This contradicts with the conclusion that the reason for higher activity is due to the*

regeneration of boron hydroxyls. ”

Reply: After considering the adsorption of propane and desorption of gas phase free radicals, the thermodynamic barrier of G to H is 0.62 eV. The net barrier of the regeneration process is 0.93 eV. Compared with the dissociation energy of O-H bond over non-metal boron based catalysts reported in other paper,^{5,6} the O-H cleavage and regeneration on the Ni@BO_x model are much lower than the cleavage barrier of non-metallic BN (2.06 eV) and boron oxides (2.31 eV). As a result, the modification of the geometric and electronic structure of BO_x overlayer by encapsulated Ni NPs is the reason for the remarkable activity of the Ni@BO_x/BN catalyst.

Comment 5. *“How stable is the propyl spectator in the surface? Is this verified from in situ spectroscopy? What is the barrier of the second C-H activation to form propene on the surface? This mechanism would not hold if it is easier for propyl to just continue to react and leave as propene, and the surface B-O* radical would consequently no longer be present. ”*

Reply: For ODHP reaction over boron-based catalysts, researchers have done a lot of work to explore its mechanism, but it is still controversial due to the complicated reaction network. We have observed some features corresponding to the B-C bond in the in-situ IR spectra at 1200 to 1090 cm⁻¹,⁷ which is the evidence of the existence of propyl species on the surface of catalysts (Figure R4). We are not able to find the transition state for the cleavage of the second C-H bond from the B-C₃H₇. Therefore, we terminated the B atom with the propyl group in order to focus on the conversion of the boron hydroxyl, which is a reasonable simplification under low conversion. The barrier of the recombination of the B-O-B bond (C to A) is also much higher than the reaction cycles. Therefore, we think the recombination is not an important route under working condition.

Figure R4. The observation of B-C vibration by in-situ DRIFTS spectroscopy

If taken the product water into consideration (high conversion), we found that the B-C₃H₇ is not stable as it will react with the product water to form another hydroxyl (Figure R5, thermodynamically highly favorable with free energy change of -1.09 eV). Some papers^{8,9} also reported that the B-C₃H₇ can react with gas phase C₃H₇O· radical to generate ·C₃H₇ radicals. While the B-O-C₃H₇ will further decompose to generate propene and B-OH.⁸ In other word, if we consider the reaction network containing reactant and product water and other gas phase radicals, the most important step will still be the cleavage and regeneration of the O-H bond of boron hydroxyl.

Figure R5. The thermodynamic calculation of the reaction between B-C₃H₇ species with water

We appreciate the reviewer's insightful comment and will try to evaluate the reaction network in the future studies. However, it can be anticipated that the work will be extremely time- and resource-consuming as the multiple layer $\text{BO}_x/\text{Ni}(111)$ is a very large system and numerous elementary steps should be considered.

Referee #3

This contribution investigates the catalytic performance of boron oxide formed over metallic Ni nanoparticles in the oxidative dehydrogenation of propane (ODHP) by combined using of experimental and theoretical approaches. The authors successfully synthesize $\text{Ni}@\text{BO}_x/\text{BN}$ by using Ni-boron nitride hybrid materials after treatment at 800 °C in a CO_2 flow and this catalyst shows a remarkable low-temperature activity for ODHP compared to the conventional boron-based catalyst. In my view, this study provides another route for preparing a boron-based catalyst with improved activity and the catalytic properties have been well characterized. I have however some serious concerns as outlined below:

Comment 1. *“Although the Ni species encapsulated by BO_x play a key role in the $\text{Ni}@\text{BO}_x/\text{BN}$ catalyst for ODHP, unlike the case of the Ni species, there is a lack of elucidation for understanding and supporting the role of B (e.g., BO_x) species in the catalyst. Indeed, the catalysts with and without BO_x species (i.e., $\text{Ni}/\text{BN}-\text{W}$ and $\text{Ni}@\text{BO}_x/\text{BN}$) exhibit the very different catalytic behaviors for ODHP. Thus, the details of the physicochemical properties of BO_x in $\text{Ni}@\text{BO}_x/\text{BN}$ with the experimental data should be described more precisely.”*

Reply: Thanks for the suggestions of the reviewer. In this manuscript, we demonstrated that the BO_x overlayer is the active site for the excellent ODHP performance of $\text{Ni}@\text{BO}_x/\text{BN}$ catalyst and explained the function of the overlayer. We agree that the characterization of the structure of active site is important to obtain in-depth understandings on the catalytic performances. As a result, we tried to study the $\text{Ni}@\text{BO}_x/\text{BN}$ catalysts using various spectroscopic, diffraction and microscopy imaging methods and the results are shown in the manuscript and supporting information. From the characterization results, we confirm that the overlayer is

composed with amorphous boron oxide and the thickness of the shell is around 0.5~2 nm. However, the surface area of BO_x overlayer is only 2% of the exposing surface area of the $\text{Ni@BO}_x/\text{BN}$ catalyst based on the chemisorption results. Considering thickness of the BO_x overlayer is 0.5-2 nm (much smaller than the BN support), it is reasonable to anticipate that the molar percentage of BO_x active species are is less than 0.1% in the bulk phase. Therefore, it is difficult to study the properties of BO_x overlayer in detail using available bulk and surface sensitive techniques, especially when the B and O elements may also exist in large amount in the BN support, due to the insufficient sensitivity of diffraction and spectroscopic methods. The advanced electron microscopy coupling with chemical sensitive electron spectroscopic techniques and in-situ sample environment may provide the required local sensitivity to study the nature of the BO_x overlayer under working condition. We will try to seek collaboration to perform the in-situ characterization in the future.

Comment 2. “ As mentioned in this manuscript, the $\text{Ni@BO}_x/\text{BN}$ catalyst treated at 800 °C in a CO_2 flow shows increased surface oxygen species. If the surface oxygen species formed by CO_2 is attributed to BO_x , which influences the catalyst activity, it is recommended to compare the catalyst properties treated under CO_2 and O_2 , respectively. This will help to better understand the effect of BO_x species on this reaction, and whether it can be controlled during the catalyst preparation step.”

Reply: Thanks for your suggestion, we prepared the catalyst using the same method to $\text{Ni@BO}_x/\text{BN}$ and replaced CO_2 with O_2 flow (The catalyst is denoted as NiO/BN). The XRD of the NiO/BN only exhibits the diffraction peaks of NiO phase, suggesting all the Ni species are oxidized after the pretreatment (Figure R6). The previous paper has reported that the B_2O_3 layer cannot form on the NiO particles under this condition (O_2 flow at 800 °C for 2 h) due to the oxidation of Ni core.¹⁰ The catalytic performance for ODHP reaction of $\text{Ni@BO}_x/\text{BN}$, BN and NiO/BN catalysts were shown together in Fig. R7. It is clear that the NiO/BN is the worst catalyst among the tested samples. The low temperature conversion of propane is even smaller than the bare BN support. Therefore, NiO does not contribute to the improved performance of ODHP. Meanwhile, it also

suggests that the sub-surface Ni boosted boron oxide layer plays the major role for the ODHP reaction to propene in our catalyst system.

Figure. R6 The XRD pattern of NiO/BN catalyst

Figure R7. The catalytic activity of different catalyst

Comment 3. “As shown in Figures 2 and 3, the Ni@BO_x/BN catalyst shows the better performance than the conventional boron-based catalyst. However, the catalyst performance in ODHP can be over- or underestimated due to the heat and mass transfer limitations (Org. Process Res. Dev. 2018, 22, 1644–1652). Therefore, more in-depth discussions of the catalytic results are required to clearly explain the transfer limitation in the Ni@BO_x/BN catalyst system in comparison with the conventional one.”

Reply: According to your suggestion, we have explored the effect of transfer limitation on both Ni@BO_x/BN and BN catalysts and the results are shown in Fig. R8.

Fig.R8(a) shows the reaction rates of catalysts at different sizes of catalyst grains. The reaction rate of catalysts with pellet sizes of 180 - 425 μm is approximately constant

for both catalysts, indicating that the intraparticle gradients play a minor role in the activity. It also shows that the reaction rate of Ni@BO_x/BN catalyst is higher than that of BN at any particle size range.

To understand the effect of external mass transfer on the reaction rate, the weight of BN and Ni@BO_x/BN catalysts were changed from 25, 50 to 100 mg in the ODHP reaction. The total gas flow was adjusted to obtain different W/F values. Fig.R8b displays the conversion of propane as a function of W/F value of the catalysts. With the increasing space velocity, the catalytic reactivity decreases for both BN and Ni@BO_x/BN catalysts. Meanwhile, we also performed the reaction in the reactor with different inner diameter (6 mm and 10 mm, see Fig. R8c). At the same space velocity, the reaction rate in the 10 mm (ID) tube is much higher than that in the 6 mm tube, which is due to the large contribution of gas free radical pathways. This result also confirms that accelerating the radical initialization is an effective way to promote the activity of ODHP catalysts.

The Ni@BO_x/BN catalyst shows significant activity advantages over the BN counterparts in all these experiments. Therefore, the subsurface Ni promotion effect on the BO_x overlayer proposed in this manuscript is highly efficient in the ODHP reaction.

Figure. R8. (a) Propane consumption rate as a function particle size (reaction conditions: 440 °C,

V/V/V of C₃H₈/O₂/N₂ = 1/1.5/3.5, 0.1 Mpa), (b) Propane conversion as a function of W_{cat}/F_o for various BN masses (reaction conditions: 440 °C, 0.1 Mpa, WHSV =14400 L·kg⁻¹·h⁻¹) and (c) Propane conversion as a function of W_{cat}/F_o for various ID tube (reaction conditions: 440 °C, 0.1 Mpa, WHSV =14400 L·kg⁻¹·h⁻¹)

Comment 4. *“Figure 4 summarizes the computational results. Step A => B => C is considered an initiation step, generating the active species in the catalytic cycle. This step is very questionable. Subsequently, the authors let adsorbed O2 abstract an H-atom from BO-H to form both BO* and HOO* radicals. This step is problematic. The difference in BO-H and OO-H bond dissociation energy is likely much larger than the 0.25 eV (5.8 kcal/mol) predicted. The authors did not benchmark their computational level for such open-shell species. Assuming O2 would be able to abstract that H-atom, why would it not directly abstract H-atoms from propane as the C-H BDE in propane is weaker than XO-H bonds? This part of the work is pretty shaky and needs to be addressed properly.”*

Reply: Thank for your suggestions. The mechanism of the ODHP reaction is very complicated, which involves both surface reaction and gas phase radical processes. Therefore, the aim of the DFT calculation in this manuscript is to understand the promotion effect of subsurface metallic Ni particle, rather than to clarify the whole reaction networks. As a result, we make appropriate simplification in the theoretical calculations and mainly focus on the chemical variation of the active site in the radical initialization and regeneration steps.

Step A => B => C is the activation process, in which the boron hydroxyl active species generate. Although energy barrier from A to C is relatively high, it is still able to proceed for a reaction performed above 400 °C. Compared with the steps for the catalytic cycles, the recombination of the B and O atoms back into BO_x shell is more difficult due to the relatively higher barrier. Therefore, the pre-activation step is reasonable. The IR spectra demonstrate that large amount of B-OH species formed after treating the catalyst with reaction atmosphere, which serves as a solid experimental evidence for the existence of the activation process.

For the reported non-metallic boron-based catalysts, it is hard for adsorbed O_2 to abstract an H-atom from BO-H to form both $BO\cdot$ and $\cdot OOH$ radicals, which is the rate-limiting step in ODHP reaction over nonmetallic boron catalysts.⁶ Even though, in the theoretical studies in literature, the C-H abstract is still performed by the activated oxygen radicals instead of dioxygen. On the Ni@BO_x/BN catalyst, the function of the subsurface Ni is to active the O-H bond of the boron hydroxyl group and reduces the BDE of BO-H bond approaching the BDE of C-H bond. Therefore, only the barrier of the H abstraction from the boron hydroxyl by O_2 is significantly reduced. While, the ability for the H abstraction of gas phase O_2 from propane is not affected. As a result, the kinetic barrier of the initialization step has been reduced significantly. The functionality of the subsurface Ni is to weaken the O-H bond and accelerate the formation of $\cdot OOH$ radicals.

References

- 1 Grant, J. T. *et al.* Selective oxidative dehydrogenation of propane to propene using boron nitride catalysts. *Science* **354**, 1570-1573(2016).
- 2 Zhang, Z., Jimenez-Izal, E., Hermans, I. & Alexandrova, A. N. Dynamic Phase Diagram of Catalytic Surface of Hexagonal Boron Nitride under Conditions of Oxidative Dehydrogenation of Propane. *J. Phys. Chem. Lett.* **10**, 20-25, (2019).
- 3 Dorn, R. W. *et al.* Structure Determination of Boron-Based Oxidative Dehydrogenation Heterogeneous Catalysts With Ultrahigh Field 35.2 T 11B Solid-State NMR Spectroscopy. *ACS Catalysis* **10**, 13852-13866 (2020).
- 4 Dong, J. *et al.* Reaction-Induced Strong Metal–Support Interactions between Metals and Inert Boron Nitride Nanosheets. *J. Am. Chem. Soc.* **142** (2020).
- 5 Shi, L. *et al.* Selective oxidative dehydrogenation of ethane to ethylene over a hydroxylated boron nitride catalyst. *Chin. J. Catal.* **38**, 389-395 (2017).
- 6 Venegas, J. M. *et al.* Why Boron Nitride is such a Selective Catalyst for the Oxidative Dehydrogenation of Propane. *Angew. Chem. Int. Ed.* **59**, 16527-16535 (2020).
- 7 He, D. *et al.* Relief of the internal stress in ternary boron carbonitride thin films. *Appl. Surf. Sci.* **191**, 338-343 (2002).
- 8 Liu, Z., Lu, W.-D., Wang, D. & Lu, A.-H. Interplay of On- and Off-Surface Processes in the B₂O₃-Catalyzed Oxidative Dehydrogenation of Propane: A DFT Study. *J. Phys. Chem. C* **125**, 24930-24944 (2021).
- 9 Liu, Z. *et al.* Understanding the Unique Antioxidation Property of Boron-Based Catalysts during Oxidative Dehydrogenation of Alkanes. *J. Phys. Chem. Lett.* **12**, 8770-8776 (2021).
- 10 Song, T. *et al.* Oxidative Strong Metal–Support Interactions between Metals and Inert Boron Nitride. *J. Phys. Chem. Lett.* **12**, 4187-4194 (2021).

REVIEWERS' COMMENTS

Reviewer #1 (Remarks to the Author):

The authors revised the manuscript and answered all my questions. Not all of the reviewers' questions were answered satisfactorily, in my opinion. As an example, I mention here only the question about external mass transport limitation. I cannot understand the method used to exclude this. W/F should have been kept constant.

I also doubt the robustness of the core-shell structure. I point out that the melting point of B₂O₃ is 450°C. Thin films certainly melt at even lower temperatures. Thus, the model applied in the DFT calculations may not be relevant either.

In general, the manuscript can of course be published despite these objections and thus be discussed in the community.

I still have one remark to the sentence in the abstract:

„Oxidative dehydrogenation of propane is a potential ground-breaking technology for the preparation of propene.” This is perhaps not be the case, because in oxidative dehydration we always lose all valuable hydrogen through the formation of water. Only an economic analysis of the process can show whether ODH makes sense or not in view of an actual market situation.

Reviewer #2 (Remarks to the Author):

Authors have made some revisions in response to the comments. However, the mechanistic portion of the study still remains a point of contention and the concerns regarding the mechanism and intermediates were not fully addressed.

Reviewer #3 (Remarks to the Author):

The authors provided extensive revisions and most questions/concerns are adequately addressed.

Response to the referees:

We thank the referees for the insightful and constructive comments and suggestions in order to help us further improve the manuscript. We have addressed all the comments point-by-point and revised the manuscript accordingly. In this response letter, comments from the referees are summarized in black typeface with the original comments quoted in *italic*, and our responses are in blue typeface. All major changes have been highlighted with yellow background in the main text. According to the editorial policies and formatting requirements, we have revised our manuscript and the modifications are highlighted with a yellow background. A detailed list of changes in the manuscript is also provided.

A list of main changes we have made:

1. The DFT calculation (State F to H) was modified to explain the mechanism clearly. A parallel hydrogen abstract reaction of the BO· radical with H₂O₂ intermediate is added (F to G' to H').

A point-to-point response to the comments

Referee #1 The reviewer 1 considers that our research can be published as we have answered all his/her questions. However, there is still some confusions for reviewer 1 about our responses to other reviewers' questions.

Comment 1. *“The authors revised the manuscript and answered all my questions. Not all of the reviewers' questions were answered satisfactorily, in my opinion. As an example, I mention here only the question about external mass transport limitation. I cannot understand the method used to exclude this. W/F should have been kept constant. I also doubt the robustness of the core-shell structure. I point out that the melting point of B₂O₃ is 450°C. Thin films certainly melt at even lower temperatures. Thus, the model applied in the DFT calculations may not be relevant either.*

In general, the manuscript can of course be published despite these objections and thus

be discussed in the community.

I still have one remark to the sentence in the abstract:

„Oxidative dehydrogenation of propane is a potential ground-breaking technology for the preparation of propene.” This is perhaps not be the case, because in oxidative dehydration we always lose all valuable hydrogen through the formation of water. Only an economic analysis of the process can show whether ODH makes sense or not in view of an actual market situation.”

Reply: We appreciate the reviewer’s positive recommendation and valuable comments. We will response to the concerns point by point below.

To clarify the influence of external and internal transport on the catalytic performances, we performed systematic investigation according to the suggested reference (Org. Process Res. Dev. 2018, 22, 1644–1652). To evaluate the external transfer limitation, there are two classic methods. The first one is mentioned by reviewer 1, in which the dependence of conversion with the flow rate is evaluated under constant W/F. When the conversion is independent with the flow rate the external transport limitation is excluded. While another method is to measure the dependence of conversion with the changing W/F over different catalyst bed with varied mass (we used this method). When there is no external transport limitation, the curves will overlap with each other. Otherwise, the more catalyst used the higher conversion of substrate will be obtained. These two methods are valid for the surface catalytic processes. However, the ODHP reaction is a complex reaction in which gas phase radical reaction contributes significantly to the conversion of propane. As a result, we observed that the catalyst bed with less amount of catalyst is more active due to the lower possibility of radical termination on the catalyst (Figure S16b). By comparing the performance of BN and Ni@BO_x/BN catalysts under various conditions, the conclusion that Ni@BO_x/BN is more active than the BN is reliable.

In this manuscript, we designed a number of experiments and characterizations to demonstrate the stability of core-shell structure. The electron microscopic images of the spent catalysts proved the boron oxide layer is robust under relatively long-time

reaction conditions. The reason for the stability of core-shell structure is because it is under dynamic equilibrium condition. Indeed, the BO_x shell can certainly be leached in the high temperature steam (this is why we can selectively remove the shell by treating the catalyst with hot water). However, the etching of the BN support, the generation of BO_x fragments and the SMSI-driven encapsulation of the metal NPs occurred simultaneously under working condition¹. As a result, the shell is self-healing in the ODHP reaction and there will always be a dense shell of BO_x on the surface of large Ni NPs. The DFT calculation is performed on an idealized model based on the structural understandings, which is reasonable in our opinions.

We agree with the reviewer's statement that "Only an economic analysis of the process can show whether ODH makes sense or not in view of an actual market situation ". The development of low temperature ODHP catalyst with significantly enhanced activity in this paper will provide more options for the process engineering. The discovery of the subsurface metal NPs working as the promoters of boron based catalytic materials is scientifically important and has not been revealed in the previous studies. Therefore, the abstract is revised carefully to focus on the scientific values instead of the technical ones. The sentence is replaced by "Oxidative dehydrogenation of propane is a promising technology for the preparation of propene." in the revised manuscript.

Referee #2 The reviewer 1 considers that we have made some revisions in response to the comments. However, the mechanistic portion of the study still remains a point of contention and the concerns regarding the mechanism and intermediates were not fully addressed.

Reply: Thanks for your valuable comments and according to your question, we have further revised the DFT calculation. The mechanism of ODHP reaction over boron-based catalysts is compiled and still controversial as it involves both surface and gas radical reactions. Therefore, we focused on the radical initialization enhancement of Ni@ BO_x /BN catalyst in the ODHP reaction based on the experimental results. Therefore, the DFT calculation focused on the catalytic cycles of $\text{BO-H} \leftrightarrow \text{BO}\cdot$ on the $\text{B}_2\text{O}_3/\text{Ni}(111)$ model surface. In the previous studies, we have demonstrated the oxygen assisted O-H dissociation of the boron hydroxyl can be effectively proceed. However,

only C₃H₈ is considered as the hydrogen source for the H abstract reaction of the surface BO· radicals (State F to H). Therefore, in the revised paper, we also considered the potential hydrogen abstract from other gas phase intermediates H₂O₂.^{2,3} We also provide the total conversion pathways of the ODHP of corresponding manner of O-H bond regeneration in the supporting information (Table S8). It can be seen that the thermodynamic advantage of the BO· + H₂O₂ → BOH + ·OOH is larger the H-abtract from adsorbed C₃H₈, suggesting most of the propane may be converted via the gas phase conversion pathway. The result of the DFT calculation agree with the kinetic studies and DRIFTS experimental results of this paper and agree with the recent mechanism studies that the gas phase radical reaction is major reaction route of ODHP reaction.^{2,3}

Reviewer #3: The reviewer 3 considers that the questions/concerns are have been adequately addressed.

Reply: Thank for your positive comments to help us improve the manuscript.

References

- 1 Sheng, J. *et al.* Preparation of oxygen reactivity-tuned FeO_x/BN catalyst for selectively oxidative dehydrogenation of ethylbenzene to styrene. *Appl. Catal., B* **305**, 121070 (2022).
- 2 Venegas, J. M. *et al.* Why Boron Nitride is such a Selective Catalyst for the Oxidative Dehydrogenation of Propane. *Angew. Chem. Int. Ed.* **59**, 16527-16535 (2020).
- 3 Liu, Z., Lu, W.-D., Wang, D. & Lu, A.-H. Interplay of On- and Off-Surface Processes in the B₂O₃-Catalyzed Oxidative Dehydrogenation of Propane: A DFT Study. *J. Phys. Chem. C* **125**, 24930-24944 (2021).